# RNA virus attenuation by codon pair deoptimisation is an artefact of increases in CpG/UpA dinucleotide frequencies

**Fiona Tulloch[1], Nicky J Atkinson[2], David J Evans[3], Martin D Ryan[1], Peter Simmonds[2]\***

[1]School of Biology, University of St Andrews, St Andrews, United Kingdom; [2]Infection and Immunity Division, Roslin Institute, University of Edinburgh, Edinburgh, United Kingdom; [3]School of Life Sciences, University of Warwick, Coventry, United Kingdom

**Abstract** Mutating RNA virus genomes to alter codon pair (CP) frequencies and reduce translation efficiency has been advocated as a method to generate safe, attenuated virus vaccines. However, selection for disfavoured CPs leads to unintended increases in CpG and UpA dinucleotide frequencies that also attenuate replication. We designed and phenotypically characterised mutants of the picornavirus, echovirus 7, in which these parameters were independently varied to determine which most influenced virus replication. CpG and UpA dinucleotide frequencies primarily influenced virus replication ability while no fitness differences were observed between mutants with different CP usage where dinucleotide frequencies were kept constant. Contrastingly, translation efficiency was unaffected by either CP usage or dinucleotide frequencies. This mechanistic insight is critical for future rational design of live virus vaccines and their safety evaluation; attenuation is mediated through enhanced innate immune responses to viruses with elevated CpG/UpA dinucleotide frequencies rather the viruses themselves being intrinsically defective.

**\*For correspondence:** Peter.Simmonds@ed.ac.uk

**Competing interests:** The authors declare that no competing interests exist.

**Reviewing editor**: Stephen P Goff, Howard Hughes Medical Institute, Columbia University, United States

## Introduction

Protein encoding regions of all organisms, eukaryotic, bacterial and viral, are subject to a number of functional constraints in addition to coding capacity, many of which contribute to regulation of translation. These include the widely reported biases in the relative frequencies of codons encoding the same amino acid (*Bennetzen and Hall, 1982*; *Sharp et al., 2005*; *Wu et al., 2010*) which in some organisms represents optimisation of the coding sequence for specific tRNAs, elongation rates and translation accuracy (reviewed in *Gingold and Pilpel, 2011*). There are, in addition, consistent under- and over-representations of codon pairs (CPs) in all organisms (*Yarus and Folley, 1985*; *Gutman and Hatfield, 1989*; *Boycheva et al., 2003*; *Moura et al., 2005*; *Tats et al., 2008*) that have been proposed to influence gene expression through alterations in translation efficiency.

Because of its potential effect on gene expression, altering CP frequencies towards those that are disfavoured in their hosts has recently been advocated as a novel strategy to reduce RNA virus replication (*Coleman et al., 2008*; *Wimmer et al., 2009*; *Mueller et al., 2010*; *Martrus et al., 2013*; *Yang et al., 2013*; *Le Nouen et al., 2014*; *Ni et al., 2014*). This procedure potentially provides the means to produce a new generation of safer, non-reverting, live attenuated vaccines. Classically, virus genomes have been empirically attenuated by serial passage in tissue-culture leading to the accumulation of mutations. This lengthy, stochastic process produced attenuated virus vaccine strains which have produced major effects on human (eg., poliovirus—Sabin vaccines) and animal health (eg., eradication of Rinderpest using the Plowright vaccine). However, reversion to virulence by back-mutation of characteristically a small number of key, attenuating, mutations is a well-known problem. Novel strategies by which synonymous coding changes are introduced to modify codon usage (*Mueller et al., 2010*) (*Martrus et al., 2013*; *Yang et al., 2013*; *Le Nouen et al., 2014*; *Ni et al., 2014*) have the advantage

**eLife digest** Viruses cause a number of diseases in humans, such as measles or polio, that can be prevented by vaccines. Some vaccines contain whole *viruses* that have been weakened or modified so that they do not cause illness. In response to infection with these weakened viruses, the immune system creates cells that can 'recognise', and protect against, the disease-causing forms of the virus if these are encountered later.

Directly altering the genetic material of viruses has been suggested as a safe way of weakening them for use in vaccines. Viruses store their genetic material in strands of either DNA or RNA, which are each made up of building blocks called nucleotides. Each nucleotide is commonly represented by a single letter—for RNA, these are A, C, G and U. These letters are read in blocks of three, called codons, when the RNA sequence is translated to make proteins.

Some codons perform the same tasks. However, in many organisms, a 'codon bias' exists, where one codon is used more often than the others that could perform the same role. Certain codons also tend to be found next to each other while some codon pair combinations are avoided; this is called 'codon pair bias'. Separately, some individual nucleotides are more likely to occur in certain pairs than others. The nucleotide pairs CG and UA, for example, generally occur less often than expected in the RNA of viruses, as well as in the genomes of the animals and plants they infect.

Currently, much research is focusing on developing weakened viruses for vaccines by introducing unfavourable host codon pairs into viral RNA, as this has been suggested to reduce how efficiently virus RNA sequences are translated into proteins. However, Tulloch et al. noticed that introducing these unfavourable codon pairs also increases the number of CG or UA nucleotide pairs present in the RNA of the virus. It is therefore unclear whether increasing the number of unfavoured codon pairs or unfavoured nucleotide pairs produces the weakening effect.

Tulloch et al. took a virus that infects the human gut, called echovirus 7, and created two types of mutant virus. One set of viruses had altered codon pair frequencies but the overall number of CG or UA nucleotide pairs was kept constant. In the other set, codon pair frequencies were kept the same but CG and UA frequencies were changed. Tulloch et al. found that altering codon pair frequencies did not affect virus replication or fitness. In contrast, increasing CG or UA nucleotide frequencies weakened the virus. Other work by the researchers involved in this study has shown that this impaired replication arises through the viruses being more readily targeted by the immune response of the invaded cell; the viruses themselves are not replication-defective.

Engineering viruses with precisely tuned replication properties is a breakthrough in the quest for safer virus vaccines, and provides useful tools for investigating virus–host interactions. The dramatic effect on viral replication of altering nucleotide pair frequencies has revealed the existence of new pathways that defend cells against virus infections, knowledge that could be exploited to rationally design vaccines.

that the resulting virus attenuation is dependent on a large number of mutations each of which only slightly reduce replicative fitness, but taken together produce significant attenuation with greatly enhanced genetic stability.

As one of the first examples, *Coleman et al. (2008)* generated synthetic poliovirus capsid gene sequences containing codon pairs that were specifically disfavoured in human coding sequences. These CP de-optimised sequences were inserted into an infectious cDNA clone of poliovirus. Virus generated from these mutants showed a remarkably attenuated replication phenotype attributed by the authors to impaired translation efficiency. Codon pair de-optimisation (CPD) has since been developed as a strategy for the production of a wide range of other live attenuated virus vaccines including influenza A virus (IAV), porcine reproductive and respiratory syndrome virus (PRRSV), human immunodeficiency virus type 1 (HIV-1) and respiratory syncytial virus (*Mueller et al., 2010*; *Martrus et al., 2013*; *Yang et al., 2013*; *Le Nouen et al., 2014*; *Ni et al., 2014*).

While altering translation efficiency through manipulation of codon or codon pair usage may attenuate virus replication, other virus compositional features may additionally contribute to replication phenotypes. One prominent compositional abnormality among RNA and small DNA viruses infecting mammals and plants is the marked suppression of the frequencies of CpG and UpA dinucleotides

(*Karlin et al., 1994*; *Rima and McFerran, 1997*; *Simmonds et al., 2013*). The functional basis for this suppression was recently demonstrated by the marked attenuating effect of artificially increasing the numbers of CpG and UpA dinucleotides in the genome of echovirus 7 (E7; (*Atkinson et al., 2014*)). We and others (*Burns et al., 2009*) have speculated that effects of CpG/UpA frequencies on virus replication may indeed account for, at least in part, the attenuating effect of selecting disfavoured codon pairs in CPD mutant of poliovirus and other candidate attenuated vaccines. Supporting this conjecture, regression analysis of the effects of numerous compositional variables in a range of codon- and codon pair deoptimised mutants on poliovirus replication demonstrated the primary effect of CpG and UpA frequencies on replication ability rather than alterations in codon or codon pair usage (*Burns et al., 2009*).

In the current study, we have used a variety of bioinformatic analyses to investigate the relationship between dinucleotide frequencies and codon pair usage. We have subsequently designed and assessed the replication phenotypes and fitness of mutants of E7 constructed in such a way that allows effects of codon pair and dinucleotide frequency alterations to be separately altered. The findings demonstrate the primary influence of CpG and UpA frequencies on virus replication that was independent of codon pair usage and translation efficiency.

## Results

### Virus attenuation, CP and dinucleotide frequencies

Coding regions of poliovirus, IAV, PRRSV and HIV-1 have all been subjected to CP de-optimisation and effects on virus replication quantified (*Coleman et al., 2008*; *Mueller et al., 2010*; *Martrus et al., 2013*; *Yang et al., 2013*; *Ni et al., 2014*). Despite their diversity of replication and translation mechanisms, each showed a similar relationship between the extent of CPD and reduction in virus replication ability (*Table 1*). Typically, 10-fold or greater attenuation in cell culture required >12–15% replacement of WT genome with CPD sequences. It is notable, however, that for each virus, CPD

**Table 1.** Relationship between codon pair de-optimisation, CpG and UpA frequencies and virus fitness reduction

| Virus | Gene | Prop'n | WT | | | CPD | | | Replication Reduction | Ref |
|---|---|---|---|---|---|---|---|---|---|---|
| | | | CP bias | CpG | UpA | CP bias | CpG | UpA | | |
| Poliovirus | | | | | | | | | | |
| PV-X | Capsid | 14.8% | −0.03 | 0.52 | 0.75 | −0.46 | 1.34 | 1.25 | ×25 | *Coleman et al., 2008* |
| PV-XY | Capsid | 25.9% | −0.03 | 0.54 | 0.75 | −0.46 | 1.31 | 1.27 | ×400 | |
| Influenza A virus* | | | | | | | | | | |
| HA[Min] | Segs.4 | 11.4% | 0.02 | 0.43 | 0.64 | −0.42 | 1.65 | 1.11 | ×3.5 | *Mueller et al., 2010* |
| HA/NP[Min] | Segs.4,5 | 21.3% | 0.02 | 0.44 | 0.55 | −0.42 | 1.56 | 1.14 | ×14 | |
| PR8[3F] | Segs.1,4,5 | 29.1% | 0.01 | 0.43 | 0.53 | −0.41 | 1.55 | 1.07 | ×35 | |
| HIV-1 | | | | | | | | | | |
| A | *gag* | 4.6% | 0.03 | 0.47 | 1.04 | −0.43 | 1.43 | 1.25 | ×7 | *Martrus et al., 2013* |
| B | *gag* | 4.7% | 0.08 | 0 | 0.91 | −0.37 | 1.22 | 1.15 | ×3 | |
| C | *gag* | 4.8% | 0.03 | 0.31 | 1.00 | −0.38 | 1.50 | 1.09 | × 8 | |
| D | *gag* | 2.1% | −0.02 | 0 | 0.49 | −0.42 | 1.47 | 0.99 | ×1.5 | |
| PRRSV | | | | | | | | | | |
| SAVE5 | gp5 | 2.6%† | −0.06 | 0.63 | 0.73 | −0.38 | 1.37 | 1.14 | ×4‡ | *Ni et al., 2014*2 |

*Codon pair minimised sequences of IAV were not provided in (*Coleman et al., 2008*) and for the purposes of comparison these have been reconstructed in SSE. Note that the CP scores described in *Table 1* of that paper (−0.386, −0.420 and −0.421 for PB1, HA and NP respectively) are not minimum scores; these are in fact −0.533, −0.585 and −0.602. Therefore, for the purposes of comparison, CP score minimisation in the current study was targeted to the former values. Although the sequences generated by SSE were not identical to those obtained previously, they would demonstrate a similar distortion of dinucleotide frequencies to those used in the previous study (*Coleman et al., 2008*).
†Mutated region only (positions 147–542 in gp5).
‡Data from replication assay in PAM cells.

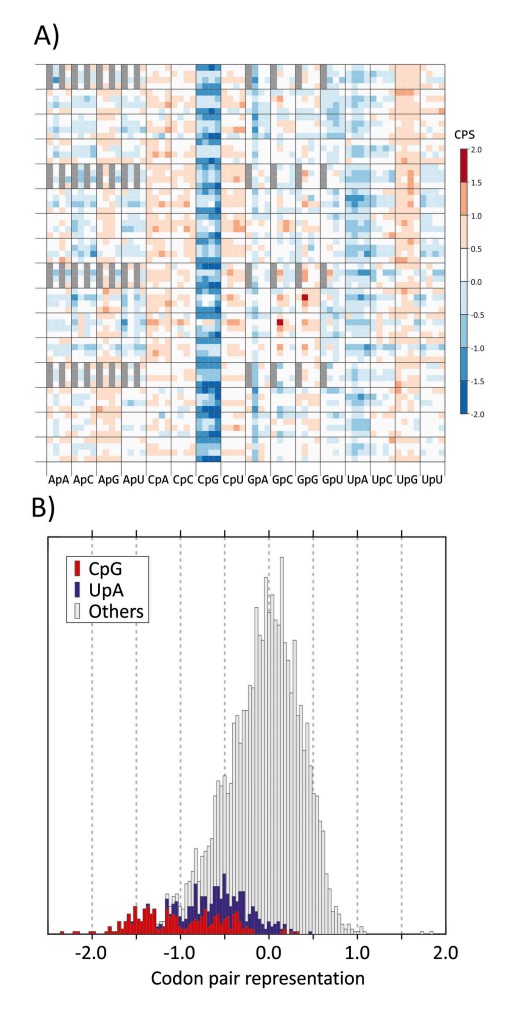

**Figure 1**. (**A**) CP usage in human coding sequences arranged in a 64 x 64 grid. CP frequencies relative to those expected from nucleotide and amino acid frequencies (CP bias) are colour coded in a heat map. The primary division on the x-axis is by identity of the 3–1 dinucleotide as annotated. Within these, further divisions within each of the 16 columns show the identity of the nucleotide at position 2 (A, C, G or U). The y-axis records nucleotides at positions 4 (4 main divisions on the y-axis), 1 (4 subdivisions of position 4) and 6 (4 subdivisions of position 1). Positions of unused codon pairs containing a 5' stop codon (translated as *|x) are shaded in grey. CP usage heat maps for *A. thaliana*, *C. elegans* and *E. coli* coding sequences are shown in *Figure 1—figure supplement 2A–2C*. (**B**) Distribution of codon pair bias scores in human coding sequences; separate labelling of the 64 codon pairs with CpG (red) or UpA (blue) across the codon junction (3–1) demonstrates their consistent under-representation based on their component nucleotide and amino acid frequencies. The distribution of codon pair scores for *A. thaliana*, *C. elegans* and *E. coli* are shown in *Figure 1. Continued on next page*

invariably increased frequencies of CpG and UpA dinucleotides (*Table 1*), typically from 0.4–0.6 to 1.4–1.6 (CpG) and from 0.5–0.8 to 1.1–1.4 (UpA) in the mutated regions.

This linkage can be accounted for at least in part by the association between CP choice and the identity of the dinucleotide between the third and first (3–1) codon positions. We analysed these parameters in coding regions of a curated dataset of over 35,170 human mRNA sequences. The representation of each of the 3904 codon pairs found in coding sequences (ie., 61 × 64) was calculated, taking into account both the nucleotide composition of the sequences and the amino acid usage as previously described (*Gutman and Hatfield, 1989*). Relative under- and over-representation of each was indicated in a heat map, with values ranging from −0.222 to +0.271 (mean 0.072). Values were plotted on x- and y-axes using values that reflected base identities at each of six positions in the codon pair (*Figure 1A*). Most of the 256 CPs with CpG at the 3–1 position (sixth main column) were markedly under-represented. There was further influence of codon pair position 6 (CP score was more suppressed if A or U) but with generally minimal and inconsistent influences of nucleotide identities at other codon positions (*Bennetzen and Hall, 1982*; *Buchan et al., 2006*; *Atkinson et al., 2014*). In the overall distribution of human codon pair representations, codon pairs containing CpG across the codon boundary distributed towards the negative tail of the distribution of CP score (*Figure 1B*) and accounted for almost all those with scores below −1.25.

CPs with UpA at the 3–1 position were also specifically under-represented in human mRNA sequences (*Figure 1A,B*), consistent with global under-representation of this dinucleotide in coding sequences from eukaryotes (*Beutler et al., 1989*; *Duan and Antezana, 2003*). The dataset additionally demonstrated over-representation of CpA and UpG dinucleotides at the 3–1 position; these are typically created by the (methylation-associated) C->T transition upstream of G (fifth and 14th main columns in *Figure 1A*) and of CpC and CpU (*Simmen, 2008*). However, with a few exceptions, such as the prominent over-representation of GCG|GCG and CCG|CCG, other codon pairs showed infrequent or minor differences in representation. The avoidance of CpG and UpA in human mRNA sequences at the 3–1 position was further manifested at other three codon position (*Bennetzen and Hall, 1982*; *Atkinson et al., 2014*); among the 61 degenerate codons, those containing CpG or UpA at these positions showed lower relative synonymous codon usage than

*Figure 1. Continued*

***Figure 1—figure supplement 3A–3C***. Correlations between codon pair scores between human coding sequences and those of *A. thaliana, C. elegans* and *E. coli* are shown in ***Figure 1—figure supplement 4***.

The following figure supplements are available for figure 1:

**Figure supplement 1**. Distribution of relative synonymous codon usage values for degenerate codons in the human genome (stop codons were excluded).

**Figure supplement 2**. CP scores of codon pairs of (**A**) *A. thaliana*, (**B**) *C. elegans* and (**C**) *E.coli* ORFeomes.

**Figure supplement 3**. Distribution of codon pair scores for other organisms-(**A**) *A thaliana*, (**B**) *C. elegans* and (**C**) *E. coli*, with separate representation of codon pairs with CpG and UpA across the codon junction.

**Figure supplement 4**. Correlation between representations of human codon pairs (x-axis) with those of other organisms-(**A**) *A thaliana*, (**B**) *C. elegans* and (**C**) *E. coli* (y-axis).

those containing other dinucleotides (***Figure 1—figure supplement 1***).

Avoidance of codon pairs with CpG at the 3–1 position was also observed in the plant genome of *A. thaliana* that also possesses a methylation-dependent suppression of CpG dinucleotides (***Figure 1—figure supplements 2A and 3***). Codon pair usage of human and plant coding sequences was indeed significantly correlated ($R^2$ = 0.146; ***Figure 1—figure supplement 4***). In contrast to plant coding sequences, no equivalent avoidance of CpG-containing codon pairs was observed in organisms with non-methylated genomes (*Caenorhabditis elegans* and *Escherichia coli*; ***Figure 1—figure supplements 2B,C,3,4***).

## Separate assessment of effects of CP and dinucleotide frequencies on virus replication

The close association between CP usage and the identity of dinucleotides at codon boundaries immediately complicates any observational assessment of the potentially separate contributions of CP bias and CpG/UpA dinucleotide frequencies on virus replication. On the one hand, it could be hypothesised that the suppression of CpG and UpA at position 3–1 in mammalian codon pairs was a simple consequence of avoiding disfavoured codon pairs. Conversely, it could be conceptualised that codon pair choice is driven in part through avoidance of specific dinucleotides. To resolve this functionally, we compared replication dynamics and relative fitness of native E7 with a series of novel mutants of E7 in which dinucleotide frequencies and CP usage were independently manipulated (***Figure 2***; ***Table 2***). To achieve this, a mutational program was developed (Sequence Mutate in the SSE package (***Simmonds, 2012***) that allowed synonymous changes to be introduced into a coding sequence to achieve a pre-specified CP score target while under constraints such as retaining CpG and UpA dinucleotide frequencies and mononucleotide composition.

The mutant, Min-E was constructed from two genome regions, together comprising 31% of the E7 genome, in which the coding sequence possessed the minimum possible CP score (−0.111) while retaining identical CpG and UpA frequencies as WT virus (CP score: −0.014; CpG: 0.525; UpA: 0.718; ***Figure 2***, ***Table 2***). Inserts with the same CP frequencies as Min-E but without dinucleotide frequency constraints (Min-U; CpG: 0.82; UpA: 0.95) or where CpG and UpA frequencies were maximised (Min-H; CpG: 1.3; UpA: 0.98) were generated similarly. The three mutants provided the opportunity to investigate effects of dinucleotide frequency differences of viral fitness without the compounding effect of CP bias. It was similarly possible to compare fitness of the mutant, Max-U, with a maximised CP score (0.320) but with similar CpG and UpA frequencies to WT with the previously described mutant, cu|cu, with minimised CpG and UpA frequencies (0 and 0.22 respectively) but a CP score marginally greater that WT (0.11; ***Figure 2***). P|P was the permuted mutant control with randomised codon order but identical coding and dinucleotide frequencies to WT sequence.

If CP usage solely determined virus replication ability, the seven mutants would be expected to display the following fitness ranking:

Max-U > cu|cu > (WT = P|P) > (Min-H = Min-U = Min-E).

Conversely, if virus fitness was determined by CpG and UpA dinucleotide frequencies, the following ranking would be expected:

cu|cu > Max-U > (WT = P|P = Min-E) > Min-U > Min-H.

These predictions were determined by generation and infectivity measurement of virus stocks corresponding to the seven mutants and comparing their relative fitness in competition and replication assays (***Figures 3,4***).

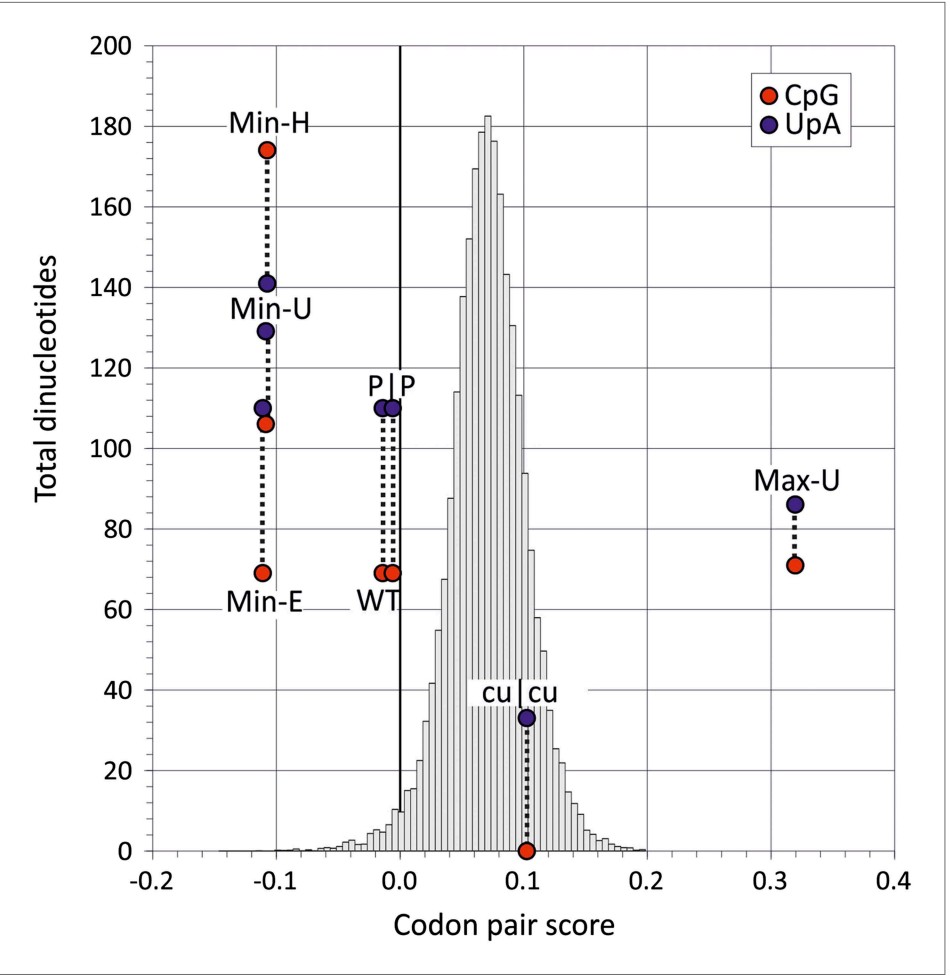

**Figure 2**. Codon pair scores and numbers of CpG and UpA dinucleotides in native (WT) and mutated regions of E7. Mean CP scores for Regions 1 and 2 combined are shown on the x-axis; the total numbers of CpG and UpA dinucleotides in each sequence are shown on the y-axis. The histogram shows CP scores for the 35,170 human mRNA sequences >200 bases in length (mean 0.072; standard deviation ±0.031).

## Replication phenotypes

Full length RNA transcribed from each E7 mutant cDNA constructs all generated infectious virus after transfection into RD cells. Stocks of virus were generated from WT and each mutant and infectivity quantified by quantal limiting dilution. To investigate replication kinetics, RD cells were infected with WT and each mutant at an MOI of 0.03 in triplicate and infectivity of supernatants measured at 8, 18 and 30 hr (*Figure 3*;). During the exponential period of replication (8 and 8 hr), Min-U and Min-H mutants showed 1 and >2 log reductions in virus replication respectively compared to WT E7. Contrastingly, the CpG/UpA-minimised mutant, cu|cu replicated to approximately 1 log higher levels that WT. Significantly for the analysis of effects of CP and dinucleotide frequencies on replication, virus titres obtained from mutant with identical (Min-E, CDLR) or similar (Max) CpG/UpA frequencies to WT were highly similar at both time points. At the last timepoint (30 hr), RD cells infected with WT, CDLR, Min-E, Max and cu|cu were entirely destroyed or almost entirely destroyed (Max-U) and showed similar residual infectivities, while those infected with Min-H showed an incomplete cytopthic effect.

Competition assays were used as a more stringent measure of fitness differences in mutants with different codon pair biases. Equal MOIs of WT and mutants were co-inoculated onto RD cells and serially passaged up to ten times. Population compositions were determined by amplification of sequences across modified regions and cleavage with restriction enzymes that differentiated WT mutant sequences from each other (*Figure 4A,B*; *Table 3*). In the examples of competition assays

**Table 2.** Composition and codon usage of E7 wt and mutant insert sequences

| Region | Sequence (Symbol) | G+C content | CpG Total* | O/E ratio†,‡ | UpA Total* | O/E ratio†,‡ | Codon Usage CAI¶ | ENc | CP Bias |
|---|---|---|---|---|---|---|---|---|---|
| 1 | Native (WT) | 47.6% | 51 (−) | 0.730 | 62 (−) | 0.742 | 0.685 | 56.5 | −0.043 |
| | Permuted (P) | 47.6% | 51 (0) | 0.730 | 2 (0) | 0.742 | 0.694 | 55.8 | −0.025 |
| | CpG/UpAL (cu) | 47.5% | 0 (−51) | 0 | 19 (−43) | 0.227 | 0.686 | 43.5 | 0.087 |
| | Max-U | 50.1% | 47 (−4) | 0.610 | 43 (−19) | 0.573 | 0.708 | 49.6 | 0.328 |
| | Min_E | 47.5% | 51 (0) | 0.736 | 62 (0) | 0.735 | 0.748 | 54.3 | −0.131 |
| | Min_U | 47.5% | 69 (+18) | 0.992 | 76 (+14) | 0.939 | 0.709 | 58.3 | −0.134 |
| | Min_H | 49.8% | 106 (+55) | 1.400 | 79 (+17) | 0.981 | 0.696 | 49.2 | −0.130 |
| 2 | Native (WT) | 47.1% | 18 (−) | 0.320 | 48 (−) | 0.695 | 0.743 | 53.2 | 0.015 |
| | Permuted (P) | 47.6% | 18 (0) | 0.320 | 48 (0) | 0.695 | 0.739 | 49.0 | 0.013 |
| | CpG/UpAL (cu) | 48.5% | 0 (−18) | 0 | 48 (0) | 0.214 | 0.739 | 47.2 | 0.118 |
| | Max-U | 46.3% | 24 (+6) | 0.440 | 43 (−3) | 0.601 | 0.750 | 46.1 | 0.311 |
| | Min-E | 45.7% | 18 (0) | 0.343 | 48 (0) | 0.657 | 0.785 | 53.3 | −0.091 |
| | Min-U | 47.4% | 37 (+19) | 0.649 | 50 (+2) | 0.738 | 0.767 | 57.6 | −0.083 |
| | Min-H | 47.8% | 68 (+50) | 1.172 | 65 (+15) | 0.970 | 0.715 | 49.7 | −0.085 |

*Total number of CpG and UpA dinucleotides in sequence. Changes in numbers between mutated and original WT sequence are indicated in parentheses.

†Ratio of observed dinucleotide frequency (O) to that expected based on mononucleotide composition (E) that is, f(CpG)/f(C) × f(G).

‡Values deliberately changed are shown in red (maximised) and blue (minimised).

¶Calculated from http://genomes.urv.es/CAIcal/ (**Puigbo et al., 2008**).

(**Figure 4A**), Max-U showed similar fitness to WT but a greater population representation at passage 10. cu|cu completely out-competed Max-U by passage 10 while in the final example, WT and Min-E showed equal fitness at passage 5 and at passage 10 (**Figure 4B**). A total of 12 pairwise comparisons were made and outcomes in terms of population representation recorded at passage 10 (**Figure 4B**; see Key). The results are internally consistent and with their replication kinetics (**Figure 3**) and indicate the following fitness ranking:

$$cu \,|\, cu > Max\text{-}U > (WT = P \,|\, P = Min\text{-}E) > Min\text{-}U > Min\text{-}H$$

Using the Spearman rank correlation test, fitness ranking was significantly associated with CpG and UpA frequencies in the insert region ($p < 0.001$) but showed no association with CP frequencies and other measures of codon usage that potentially influence translation rates, codon adaptation index (CAI) and effective number of codons (ENc) (**Table 4**). Consistently, these results demonstrate that when altered independently from CP bias, only dinucleotide frequencies were associated with replication fitness.

## Comparison of translation efficiencies

The maxim that any effects of CP frequencies on replication are mediated through its influence on translation efficiency was investigated for the mutants constructed in the study. Translation assays were evaluated in vitro to avoid effects mediated through stress response-related RNA recognition mechanisms that restrict E7 translation and subsequent replication immediately after entry (**Atkinson et al., 2014**). Viral RNA transcripts from E7 WT and mutant cDNA clones were used to program rabbit reticulocyte lysates in the presence of [$^{35}$S]-methionine. Electrophoresis of reactions after 3 hr showed translation of several bands representing cleaved and partially cleaved E7 proteins (**Figure 5**; **Figure 5—figure supplement 1**). Translation efficiencies of each of the mutant E7 transcripts were comparable to WT RNA; what variability there was between mutants (**Figure 5—figure supplement 1**) did not correlate with replication fitness ($R = −0.075$; $p > 0.5$; **Table 4**). This indicates that, at least in a whole genome context, alteration of either CP or dinucleotide frequencies had no significant effect on viral polyprotein translation and therefore cannot be attributed to the marked differences in replication phenotypes observed.

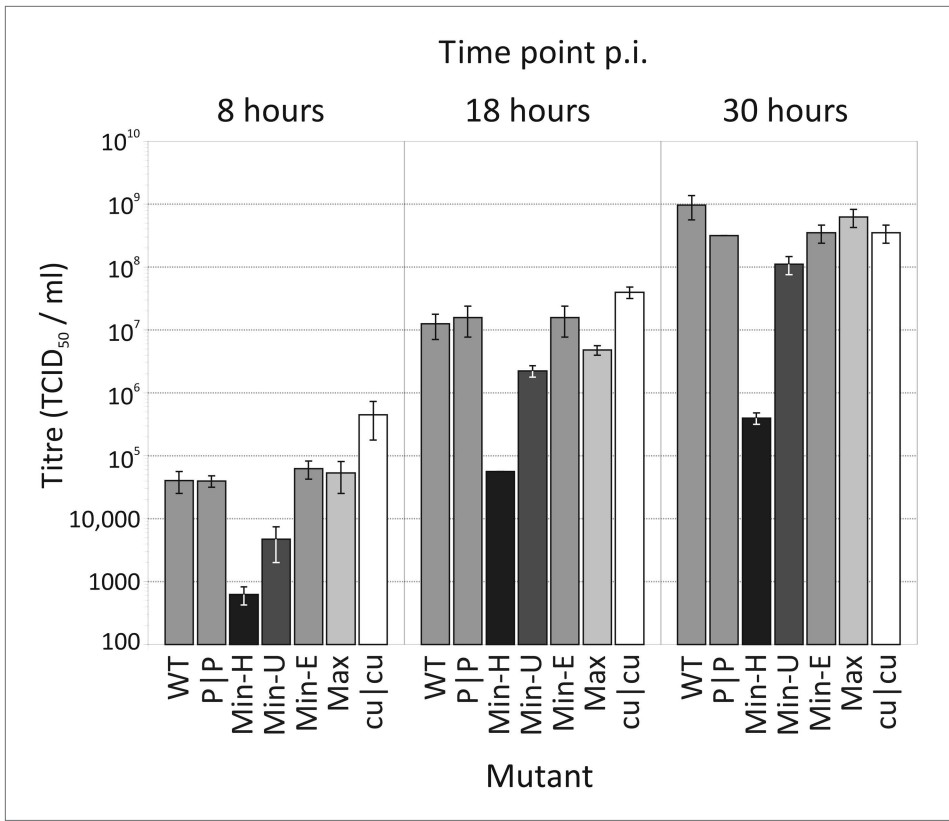

**Figure 3**. Replication of WT and mutants of E7 with altered CP and dinucleotide frequencies. Bars are shaded diagrammatically based on their relative CpG/UpA composition. RD cells were infected with E7 WT, at an MOI of 0.03 and infectious titres quantified at 8, 18 and 30 hr time points post inoculation (p.i.) by $TCID_{50}$ determination. Results are the mean of three biological replicates; error bars show standard errors of the mean.

## Discussion

This study sought to disentangle effects of codon pair usage and nucleotide frequencies in a re-examination of their effects on the replication of an RNA virus, E7. In the literature, studies have documented effects of CP de-optimisation on virus attenuation without reference to effects of this procedure on dinucleotide frequencies (*Coleman et al., 2008*; *Mueller et al., 2010*; *Martrus et al., 2013*; *Yang et al., 2013*; *Le Nouen et al., 2014*; *Ni et al., 2014*). While frequencies of both of these dinucleotides are suppressed in most classes of mammalian RNA viruses (*Rima and McFerran, 1997*), all sequences modified to select disfavoured CPs (*Coleman et al., 2008*; *Mueller et al., 2010*; *Martrus et al., 2013*; *Yang et al., 2013*; *Ni et al., 2014*) consistently elevated frequencies of CpG and UpA dinucleotides to levels to 2.5–threefold higher levels than the original native sequences (*Table 1*). As documented in other studies (*Burns et al., 2009*; *Atkinson et al., 2014*), these dinucleotide frequencies may contribute additionally to the observed attenuation of virus replication.

Through construction of mutants of E7 in which CP frequencies was altered while keeping dinucleotide frequencies constant (WT/Min-E) and conversely, generating viruses with the same or similar CP biases but different dinucleotide frequencies (eg., Min-E/Min-U/Min-U and cu|cu/WT/Max-U), we were able to separate potential influences of these compositional variables on replication phenotype and fitness. The fitness ranking derived from competition assays (cu|cu > Max > (WT = P|P = Min-E) > Min-U > Min-H) demonstrated that it was dinucleotide frequencies that significantly influenced fitness while differences in CP usage showed no detectable phenotypic effect. Moreover, if CPD were to influence virus replication then its effect would be manifested through changes in translation rate; however, no measurable differences in translation efficiency were detected between WT and CP-optimised (Max) or de-optimised (Min) template RNA sequences. These findings are broadly consistent with results

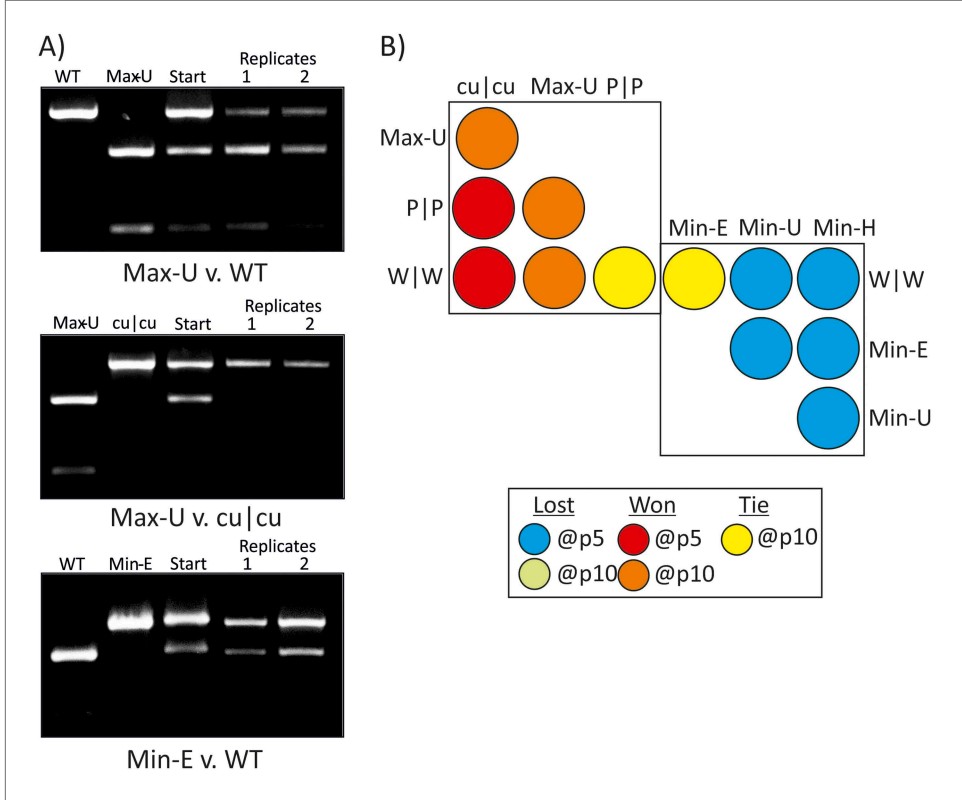

**Figure 4**. RD cells were co-infected with pairs of WT (W|W) and E7 mutants at equal MOI and the supernatant serially passaged through cells after development of CPE. RNA was isolated and the composition of each virus determined through selective restriction digests using enzymes listed in *Table 3*. (**A**) Examples of three competition assays showing cleavage patterns of individual viruses (lanes 1, 2), the starting inoculum (lane 3) and two biological replicates after 10 (panels 1, 2) or 5 (panel 3) passages in lanes 4 and 5. Results from the other competition assays are shown in *Figure 4—figure supplement 1*. (**B**) Summary of pairwise fitness comparisons of viruses with outcomes for the viruses listed in columns at passages 5 and 10 indicated by colour shading. For example, Min-E and WT showed equal fitness (yellow shading) and cu|cu outcompeted WT by passage 5 (red) and Max-U by passage 10.

The following figure supplement is available for figure 4:

**Figure supplement 1**. Competition assays between E7 mutants showing competing variants (lanes 1 and 2) andout at indicated passage number (lane 3) for each.

---

of previous translation assays of poliovirus mutants differing in CP choice in which relatively small differences in translation of PV-Min, WT and PV-Max mutants were clearly incompatible with the marked differences in their replicative ability (*Coleman et al., 2008*).

In a broader context, the finding that alterations in codon pair frequencies has no independent effect on virus replication are consistent with current understanding of the nature and driving forces behind codon pair usage in other organisms. Most importantly, there is no evidence that disfavoured CPs in eukaryotes, archaea or prokaryotes are those that specifically retard translation rates. Indeed, where specifically investigated, the opposite was observed. mRNA templates containing disfavoured codon pairs in *E. coli* were translated faster than those containing over-represented CPs (*Irwin et al., 1995*). The 16 codon pairs identified as most retarding translation of the *E. coli* his operon leader peptide gene has codon pair scores ranging from −0.94 to +0.54 and distributed around the centre of the distribution of codon pair scores (*Figure 1—Figure Supplement 3*) (*Chevance et al., 2014*). The current consensus view is that CP usage in prokaryotes is governed functionally as a means to regulate gene expression rather to maximise translation (*Folley and Yarus, 1989*; *Irwin et al., 1995*; *Boycheva et al., 2003*; *Buchan et al., 2006*).

**Table 3.** Enzymes used in selective digests for competition ASSAYS

| Virus 1 | Virus 2 | Region | Enzyme | Target |
|---------|---------|--------|--------|--------|
| W\|W | P\|P | 1 | *Spe*I | Permuted |
| W\|W | Max-U | 1 | *Sac*I | Max |
| W\|W | Min-E | 1 | *Nco*I | WT |
| W\|W | Min-U | 1 | *Nco*I | WT |
| W\|W | Min-H | 1 | *Eco*RV | WT |
| W\|W | cu\|cu | 1 | *Eco*RV | WT |
| P\|P | cu\|cu | 1 | *Spe*I | Permuted |
| Max-U | P\|P | 1 | *Spe*I | Permuted |
| Max-U | cu\|cu | 1 | *Sac*I | Max |
| Min-E | Min-U | 1 | *Cla*I | Min-U |
| Min-E | Min-H | 1 | *Eco*RV | Min-E |
| Min-U | Min-H | 1 | *Cla*I | Min-U |

In eukaryotic genomes, other factors underlie codon pair representation since coding regions in mRNA sequences and in non-transcribed genomic DNA showed similar biases in codon pair frequencies (*Moura et al., 2007*). CP frequencies must therefore be substantially determined by mutational events operating on DNA such as methylation and specific context-dependent errors during genome replication instead of any kind of optimisation or regulation of translation. The consistent under-representation of codon pairs with CpG in the 3–1 position (*Figure 1*; *Buchan et al., 2006*) in mammalian genomes indeed likely originates from DNA methylation-induced mutations in the nucleus. Our data showing similar rates of translation of Min-H and Max-U that show major differences in frequencies of CpG-containing CPs (*Figure 5*) are consistent with this interpretation.

Finally, there is no theoretical basis for the assumption that CPs are disfavoured because of their negative effects on translation efficiency and this concept runs counter to our growing understanding of the intricate mechanisms that govern gene expression. In all organisms, coding sequences differ in codon usage, match to tRNA abundances, mRNA stability and initiation sites to regulate rates and fidelity of protein expression (reviewed in *Gingold and Pilpel, 2011*). Some of the variability in CP usage observed in the three domains of life (*Moura et al., 2005*; *Sharp et al., 2005*; *Tats et al., 2008*; *Wang and Li, 2009*) likely represents aspects of that control, rather than as a means to simply maximise translation.

## Mechanism of attenuation

Understanding what limits the replication of viruses with altered CP and dinucleotide frequencies is critical in the evaluation of their broader safety as attenuated virus vaccines. The proposed mechanism in which alterations in CP bias alter translation efficiency and it is this that inhibits virus replication introduces a conceptual model in which it is the virus that is intrinsically defective. With the large number of mutations required for reversion, such viruses should be stably attenuated in whatever context they are used. However, as we have now shown, the replication defect of CPD viruses is actually mediated through alterations in dinucleotide frequencies in the genome that influence their recognition by the cell. In this alternative paradigm, viruses with elevated frequencies of CpG and UpA are not intrinsically defective but they are more readily recognised by the cell and prevented from initiating replication. Their attenuation is therefore dependent on the efficacy of the host innate immune response.

**Table 4.** Correlation between fitness ranking and sequence composition

| Variable | Spearman R | p *value*† |
|----------|-----------|-----------|
| CpG/UpA* | 1.0 | <0.001 |
| CP bias | −0.70 | 0.1 (n.s.‡) |
| CAI | −0.334 | >0.5 (n.s.) |
| ENc | 0.593 | >0.5 (n.s.) |
| G + C content | 0.075 | >0.5 (n.s.) |
| Translation efficiency | −0.074 | >0.5 (n.s.) |

*Number of CpG and UpA dinucleotides in insert region.
†From values tabulated in (*Ramsey, 1989*).
‡n.s. : not significant.

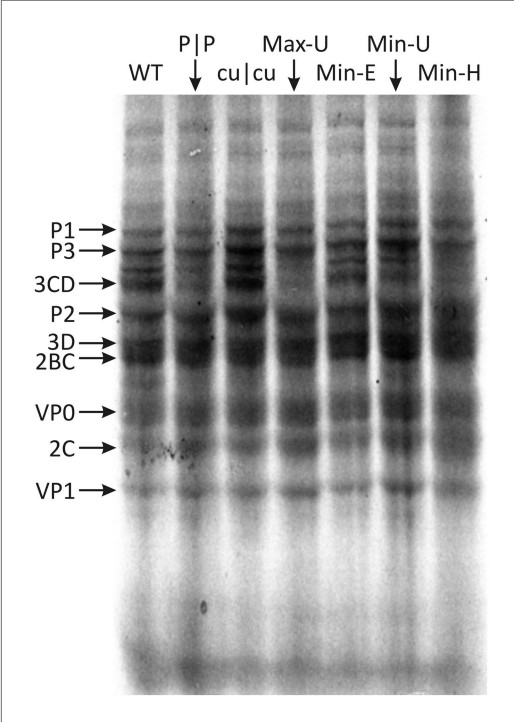

**Figure 5**. Translation of RNA templates generated from WT and mutant E7 cDNAs in a rabbit reticulocyte cell free assay. Assignments of bands to E7 proteins were based on molecular weights on SDS-PAGE. A comparison of densitometry values for viral proteins is shown in *Figure 5—figure supplement 1*.

The following figure supplement is available for figure 5:

**Figure supplement 1**. Translation efficiencies estimated by densitometry of band intensities of viral proteins translated in a rabbit reticulocyte cell free assay.

The cellular mechanisms responsible for differential recognition and response to RNA sequences with different dinucleotide composition are currently unknown. In our previous study, we obtained evidence that replication inhibition of high CpG/UpA mutants of E7 occurred shortly after cell entry and was not mediated though conventional pattern recognition receptors (*Atkinson et al., 2014*). In that study, we additionally demonstrated that it was additionally not the result of differences between high and low CpG/UpA viruses in their sensitivity to the cellular interferon response. We did observe, however, that the attenuated phenotype of high mutants could be entirely reversed by the kinase inhibitor, C16, a finding that suggests that recognition may occur through an as yet uncharacterised PKR-related component of the stress response pathway in the cell.

Both the adaptive and innate arms of the human immune system are highly polymorphic with remarkable variability in function and expression of many key components of recognition or effector proteins mediating antiviral responses (*Thomas et al., 2009*; *Everitt et al., 2012*; *Hambleton et al., 2013*; *Pothlichet and Quintana-Murci, 2013*). Although uncharacterised mechanistically, there is clearly a potential danger that pathways that restrict the replication of high CpG/UpA RNA viruses may be similarly variable in the efficacy in humans and in veterinary species with different genetic backgrounds. The attenuation of live vaccines and safety margins established for their large scale use may be similarly variable; investigation of population differences in innate cellular responses to viruses of different dinucleotide compositions is essential in the evaluation of the safety of this new generation of high CpG/UpA live attenuated vaccines.

## Materials and methods

### Cell culture and cell lines

RNA transcripts of the pT7:E7 infectious cDNA clone of the isolate Wallace (accession number AF465516) were used to generate E7 viral stocks. E7 was propagated in rhabdomyosarcoma (RD) cells using Dulbecco modified Eagle medium (DMEM) with 10% foetal calf serum (FCS), penicillin (100 U/ml) and streptomycin (100 µg/ml). All cells were maintained at 37°C with 5% $CO_2$.

### *In silico* design of CpG and UpA modified viruses

The two regions in the E7 genome used previously to investigate effects of dinucleotide frequencies on virus replication (*Atkinson et al., 2014*) were used in the current study (Region 1: 1878–3119 and 5403–6462). Previously characterised mutants comprised the CpG/UpA-low mutant cu|cu with all CpG dinucleotides and as many UpA dinucleotides possible eliminated and the permuted mutant P|P in which codon order was permuted while retaining protein coding and native dinucleotide frequencies. Further mutants (Max-U, Min-E, Min-U, Min-H) are described in the main text; sequences listed in *Supplementary file 1*.

### Bioinformatics analysis

Manipulation of dinucleotide frequencies and codon pair scores in coding sequences was performed using the program Sequence Mutate in version 1.2 of the SSE package (*Simmonds, 2012*). Reference

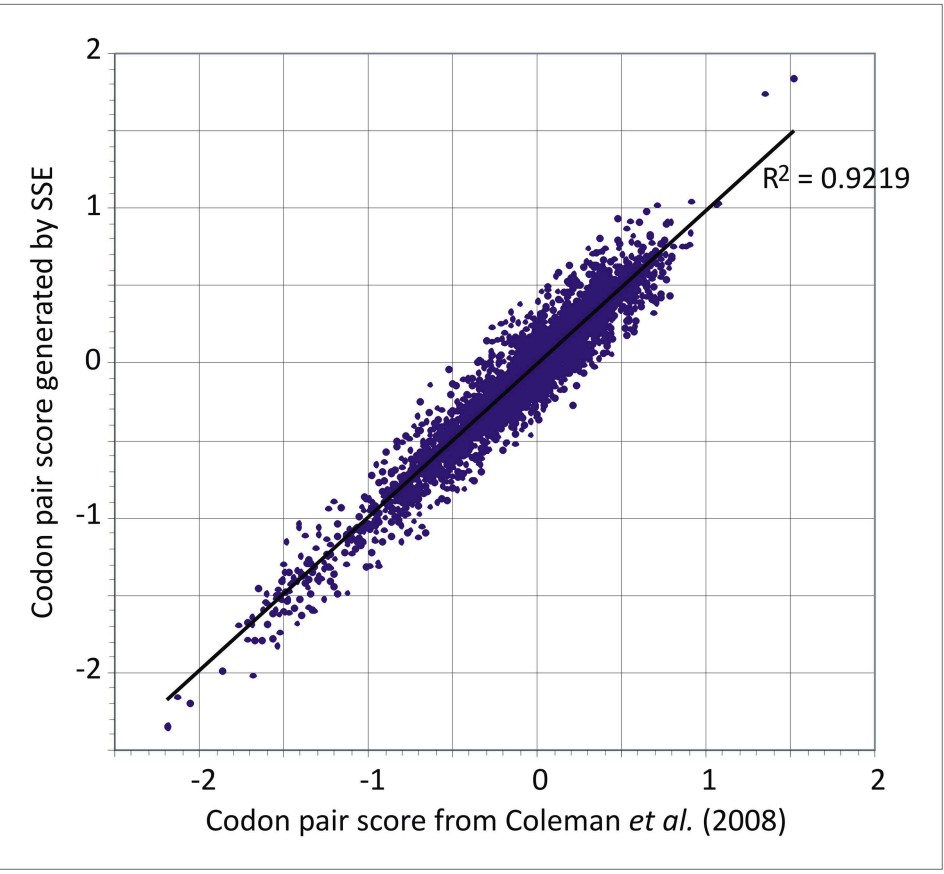

**Figure 6**. Comparison of codon pair scores generated by SSE using a dataset of 35,770 human mRNA sequences (y-axis) with those used in a previous analysis (*Coleman et al., 2008*).

datasets of human, *A. thaliana*, *C. elegans* and *E. coli* messenger RNA sequences were obtained from the Refseq database (http://www.ncbi.nlm.nih.gov/nuccore). Codon pair usage tables were generated from coding regions of each mRNA sequence datasets from the four organisms using the program Composition Scan in the SSE package (*Simmonds, 2012*). Codon pair tables generated by SSE were used to calculate CP frequencies for all mRNA sequences with coding regions >200 bases in length from each organism. These comprised 35,770 human mRNA sequences, 32,768 from *A. thaliana*, 24,093 from *C. elegans* and 4316 from *E. coli*.

The codon pair table generated by SSE from our dataset of human mRNA sequences was used in preference to that previously described (Table S2 in reference *Coleman et al., 2008*) because of the larger number of human mRNA sequences now available. The previously published dataset as presented additionally unaccountably omitted a large number of codon pairs (3586 were listed instead of the 3904 expected—61 × 64). There was however a good correlation between CP frequencies between the two datasets (*Figure 6*).

The codon adaptation index for human codon usage was calculated through the website http://genomes.urv.es/CAIcal/ (*Puigbo et al., 2008*). The effective number of codons (Enc; *Wright, 1990*) and CP usage (*Buchan et al., 2006*; *Coleman et al., 2008*) were calculated using the program Composition Scan in SSE.

## Clone construction and recovery of mutant viruses

Mutant E7 constructs with altered CP frequencies were generated from custom synthesised DNA sequences (Eurofins Genomics, Ebersberg, Germany). Mutant clones were constructed as previously described (*Atkinson et al., 2014*). All clones were sequenced over the insert regions prior to further applications. Infectious virus from each cDNA clone was recovered by transfection of RNA transcripts produced from plasmids linearised using *Not*I using a Riboprobe System-T7 in vitro transcription kit

(Promega Ltd. Southampton, UK). 100 ng of RNA was transfected into RD cells using Lipofectamine 2000 (Invitrogen, Life Technologies Ltd., Paisley, UK) according to the manufacturer's instructions. The resulting cell lysates were used to generate passage 1 stocks by re-infecting RD cells. Viral titres were determined by $TCID_{50}$ titration in RD cells.

### Replication phenotype

Multi-step growth curves for each virus were generated by infecting RD cells in triplicate in 24-well plates at an MOI of 0.03 as previously described (*Atkinson et al., 2014*). Supernatant collected at time points (8, 18 and 30 hr post-infection) were assayed for infectivity by quantal dilution. Competition assays were performed as previously described. Briefly, equal titres of virus pairs (combined MOI = 0.01) were applied simultaneously to RD cells in 25 cm² bottles. Following the development of CPE, super-natant was collected and 300 µl applied to fresh RD cells. This was continued for up to 10 passages. The results of the competition assays were determined by restriction enzyme digestion of the amplicon amplified from Region 1 by combined reverse transcription—PCR (*Atkinson et al., 2014*). Restriction enzymes used to differentiate each mutant pair are listed in *Table 3*.

### In vitro transcription and translation

RNAs were produced by in vitro T7 transcription (Riboprobe System T7, Promega) of the various cDNA plasmids, each linearised with *Not*I (Promega). Transcript RNAs were used to program nuclease-treated rabbit reticulocyte lysates (Promega) supplemented with HeLa cell S10 cytoplasmic extracts (Dundee Cell Products, Dundee, UK). Reactions were set-up as follows; 7 µl rabbit reticulocyte lysate, transcript RNA (0.25–2 µg), 0.5 µl 1 mM amino acid mix (minus methionine), 0.5 µl [$^{35}$S]-methionine (1200 Ci/mmol), 10 U RNasin Ribonuclease Inhibitor and 2.25 µl HeLa cell extract in a total volume of 12.5 µl. Reactions were incubated at 30°C for 3 hr and analysed by SDS-PAGE (4–20% Tris-Glycine, Expedeon Ltd. Cambridge, UK). Gels were exposed to film (Thermo Scientific, Basingstoke, UK) for 1–4 days at −70°C. To determine the relative density of the protein bands, densitometry was carried out on the scanned gel image using ImageJ 1.48 software (http://imagej.nih.gov/ij).

## Acknowledgements

The authors would like to thank John Nicholson and Ashley Pearson for technical assistance with the cloning and recovery of mutants of E7. The study was funded by a project grant from the Wellcome Trust (WT087628MA) and the BBSRC (BB/K003801/1 and BB/L004526/1).

## Additional information

### Funding

| Funder | Grant reference number | Author |
| --- | --- | --- |
| Wellcome Trust | WT087628MA | David J Evans, Peter Simmonds |
| Biotechnology and Biological Sciences Research Council | BB/K003801/1 | Martin D Ryan |
| Biotechnology and Biological Sciences Research Council | BB/L004526/1 | Martin D Ryan |

The funders had no role in study design, data collection and interpretation, or the decision to submit the work for publication.

### Author contributions

FT, Acquisition of data, Analysis and interpretation of data; NJA, Conception and design, Acquisition of data, Analysis and interpretation of data; DJE, MDR, PS, Conception and design, Analysis and interpretation of data, Drafting and revising the article

### Author ORCIDs

Fiona Tulloch, http://orcid.org/0000-0003-0859-2867

# Additional files

## Supplementary file
• Supplementary file 1. Nucleotide sequences of mutants used in the study. *Supplementary file 1*.docx
http://dx.doi.org/10.7488/ds/188.

## Major datasets

The following datasets were generated:

| Author(s) | Year | Dataset title | Dataset ID and/or URL | Database, license, and accessibility information |
|---|---|---|---|---|
| Simmonds P | 2014 | Composition and codon usage metrics for human mRNA | http://dx.doi.org/10.7488/ds/188 | All data is licensed under a Creative Commons Attribution 4.0 International License. |
| Simmonds P | 2014 | Composition and codon usage metrics for A. thaliana mRNA | http://dx.doi.org/10.7488/ds/188 | All data is licensed under a Creative Commons Attribution 4.0 International License. |
| Simmonds P | 2014 | Composition and codon usage metrics for C. elegans mRNA | http://dx.doi.org/10.7488/ds/188 | All data is licensed under a Creative Commons Attribution 4.0 International License. |
| Simmonds P | 2014 | Composition and codon usage metrics for E. coli mRNA | http://dx.doi.org/10.7488/ds/188 | All data is licensed under a Creative Commons Attribution 4.0 International License. |
| Simmonds P | 2014 | Codon pair usage tables for human, A. thaliana, C. elegans and E. coli | http://dx.doi.org/10.7488/ds/188 | All data is licensed under a Creative Commons Attribution 4.0 International License. |

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
