## [Decision Letter]

Thank you for sending your work entitled “RNA virus attenuation by codon pair
deoptimisation is an artefact of increases in CpG/UpA dinucleotide frequencies”
for consideration at *eLife*. Your article has been favorably evaluated
by Richard Losick (Senior editor), a Reviewing editor, and 3 reviewers, one of whom,
Raul Andino, has agreed to reveal his identity.

The Reviewing editor and the reviewers discussed their comments before we reached this
decision, and the Reviewing editor has assembled the following comments to help you
prepare a revised submission. The comments in full from the three reviewers are attached
below.

Our reviewers were in close agreement that the paper makes important points and is
topical and appropriate for *eLife*. There was also a consensus that
there were some failings in reviewing the history of this topic, and I agree that an
improved review is needed (apparently key citations are missing). This should be easily
fixed. There were some good suggestions to improve clarity. There were also requests for
more experimentation from one (the third) reviewer, which I think sound reasonable, but
I cannot evaluate how much work is involved. I would encourage the authors to consider
doing what is easy. It seems to this reviewer that the quantitation might be improved
without huge effort. We are willing to leave it to the authors to decide how much more
wet work is desired.

All told, we are supportive of publication.

Reviewer #1

This manuscript presents a careful and interesting work, and even more thoroughly
documents on the attenuating effect of dinucleotides CpG and UpA in the genomes of RNA
viruses than previous papers have. They argue, convincingly, that attenuating effects
previously ascribed to 'codon pair bias' in attenuating viruses can be
completely explained by the resulting increases in CpG and UpA dinucleotides. They do
this via a systematic analysis of the attenuating effects of altering 'codon-pair
bias' in previous papers and by carefully design experiments of their own, which
include documentation that the attenuation does not occur by reducing translational
efficiency, but by reducing the infectivity of the resulting virions.

With respect to the possible attenuation mechanism of CpG and UpA dinucleotides, the
authors discuss the possibility that it is a mechanism of the innate immune response
that is not PKR- or pattern receptor-dependent. Data relevant to this are published
separately (Atkinson et al., 2014).

The curious thing about this field, and this manuscript, is that the 'straw
man' of codon-pair bias was already destroyed by Burns et al., in 2009. These
authors were pursuing their own hypothesis, that the substitution of rare codons would
attenuate viruses. They found that this was the case, but careful mutagenesis revealed
that it was, in fact, the increase in CpG and UpA nucleotides in the RNA that accounted
for the attenuation. They showed that this effect was not due to translational
efficiency and was manifested as the increased particle-to-PFU ratio of the resulting
viruses. They reflected that bias against CpG and UpA codons might be the reason, in
fact, that codons that contain them, and codon pairs that create them, are rare to begin
with.

They noted, quite politely, that this also accounted for the effect of the highly
publicized “codon-pair bias” published by Coleman et al., 2008. They
stated: “A prominent feature of the most disfavored codon pairs is the presence
of CpG or UpA across codons. Thus, the observed CPB in poliovirus and in humans and
higher eukaryotes may be driven primarily by CpG and UpA dinucleotide suppression. In
this context, it is notable that in cassette C of the construct with the lowest fitness,
ABc_12_, within-codon CpG and UpA frequencies were maximized but the CPB
score was similar to those of higher-fitness constructs, including ABC.” Thus,
like the current manuscript, they re-analyzed the data of Coleman et al., and found it
to be completely explicable in terms of CpG and UpA dinucleotides.

This conclusion, and the published data, were clearly insufficient to prevent the design
and interpretation of several more manuscripts using 'codon pair bias' to
design attenuated viruses for vaccine and basic science purposes (Mueller et al., 2010;
Martrus, 2013; Yang, 2013; Ni, 2014).

In short, this is a well-executed, important study that extends the very clear
conclusions of the also excellent, but completely ignored, 2009 paper of Burns et al.
Therefore its publication would bring an important discussion into the limelight. Here
are my suggestions, mostly having to do with writing and scholarship:

1) That CpG and UpA dinucleotide frequency correlates with attenuation of viruses,
rather than 'codon pair bias' or 'codon de-optimization', is a
feature of these approaches rather than an artifact, in this reviewer's opinion.
The combative nature of the title is not necessary to bring this fact to light.

2) It does not detract from the quality of the present work to acknowledge more
explicitly the contributions of Burns et al. It may, however, detract from its
novelty.

3) The readers understanding of the data is compromised by the poor description of the
experiments and analysis in the figure legends. These should be rewritten in such a way
that the figures can be independently understood. For example, nowhere in the manuscript
can the metric for “CPB score” be discovered.

4) Similarly, the quantitative findings in Figure 4 are difficult to dissect. With so many changes in the genomes being
analyzed, qRT-PCR of viral competitions would be helpful. As it is, the actual data from
the competition experiments should be presented, rather than a cartoon depicting which
viruses 'won' and 'lost'.

5) The last paragraph of the Discussion uses 'data not shown' to make its
point. These data should be shown or the sentences deleted.

Reviewer #2

There has been interesting discussions on why the nucleotide composition of many human
viruses present distinct nucleotide, di-nucleotide and codon preferences. In the last
years synthetic viruses have been generated with alternative codon distributions that
show attenuated replication. However these constructs change other variables (nucleotide
or dinucleotide composition) in addition to the codon distribution and it is unclear
what of those changes affects replication. The authors address this issue by generating
mutants of the echovirus 7 in which CpG and UpA dinucleotides were varied independent of
the codon distribution. The results clearly showed that the main factor dominating
replication was dinucleotide content, and that translation efficiency was unaffected by
the two variables. The authors then argue that attenuation is mediated through innate
immune response to viruses with high CpG/UpA content.

The paper is very interesting and results are compelling. The main critiques are:

1) The topic of dinucleotide biases in RNA viruses has been explored extensively and
most references are absent.

2) The discussion on the mechanisms of attenuation and the association to the innate
immune response is not well elaborated. They mention the lack of association to
interferon response and the effect of C16. However, the results are not presented in
detail. I suggest the authors to strengthen their conclusions.

Reviewer #3

This study examines the role of nucleotide composition in RNA virus genomes. The authors
introduced a number of synonymous mutations into the EV71 genome to modify the frequency
of CpG and UpA dinucleotide (DN) or codon pair (CP). Their results support the idea that
DN frequencies determine virus replication, while neither DN nor CP affected translation
efficiency. They concluded that DN affect virus replication because the nucleotide
composition of the virus genome influences the host-cell innate response to the
virus.

The most interesting contributions of this study are: 1) they are able to identify
mutations that will affect CP without affecting CpG/UpA dinucleotide composition and
vice versa; (2) based on
this bioinformatics information they constructed mutants that change CP or DN
composition and experimentally evaluated the effect of synonymous mutation on virus
replication. Their conclusions are supported by the available results, and I think this
study represents an important contribution to the field because it seems to address a
molecular mechanism for virus genome nucleotide composition bias. However, I believe
that a more quantitative analysis of the competition experiment may be required to
determine the degree of correlation between DN composition and virus replication
fitness.

The authors choose competition assays to precisely analyze fitness between different
virus and with respect to WT. This is the correct experiment in my opinion, however I
believe the analysis of the experiment is somewhat casual and not very quantitative, and
therefore limits the value of this data. Figure 4
presents the results using a differential restriction enzyme pattern to distinguish
between the two competing viruses, but they can only determine when one virus is lost
(no longer detected by the assay). I think that it will be a lot more powerful to use
digital PCR to precisely quantify the ratio between virus genomes in the given
competition assay. This will provide parameters that can then be fit in a simple
mathematical model to determine with more accuracy the correlation between DN or CP and
fitness, which at this point seems a bit circumstantial.

Similarly, it would be desirable to improve the quality of the in vitro translation
assay and quantify protein production to determine that, in this case, there is little
correlation between translation efficiency and fitness, as this is one of the central
claims of the study.

---

## [Author Response]

Reviewer #1

*1) That CpG and UpA dinucleotide frequency correlates with attenuation of
viruses, rather than 'codon pair bias' or 'codon
de-optimization', is a feature of these approaches rather than an artifact, in
this reviewer's opinion. The combative nature of the title is not necessary to
bring this fact to light*.

As co-authors, we did spend some time discussing the title of the paper. The final
choice was in fact motivated for reasons alluded to by the reviewer: that those involved
in the codon pair programme have simply ignored the published evidence, by Burns et al.
and more recently from our lab (Atkinson et al.), that the attenuating effect was
mediated through inadvertently increasing CpG ad UpA dinucleotide frequencies rather any
effect on translation. A clear, declarative title of the paper seems required to counter
mistaken views on this. If we changed the title as the reviewer suggests, it would imply
that codon pair de-optimisation has been used deliberately as a way to increase CpG and
UpA dinucleotide frequencies. This is absolutely not the case.

That said, we would have liked to qualify that statement by stating why we make this
assertion, but the low character limit imposed on *eLife* papers titles
prevents us from doing this.

*2) It does not detract from the quality of the present work to acknowledge more
explicitly the contributions of Burns et al. It may, however, detract from its
novelty*.

We have cited that study in the original manuscript. In the Introduction of the revised
manuscript, we have described the observation made by Burns and colleagues in more
detail as requested by the reviewer.

*3) The readers understanding of the data is compromised by the poor description
of the experiments and analysis in the figure legends. These should be rewritten in
such a way that the figures can be independently understood. For example, nowhere in
the manuscript can the metric for “CPB score” be
discovered*.

We apologise for these omissions and have endeavoured to make the figure legends clearer
(more explanatory) and describe the various abbreviations more fully. We have dropped
the abbreviations CPB as it can be expressed more clearly by other wordings
(e.g*.* biased codon pair usage).

*4) Similarly, the quantitative findings in*
Figure 4
*are difficult to dissect. With so many changes in the genomes being analyzed,
qRT-PCR of viral competitions would be helpful. As it is, the actual data from the
competition experiments should be presented, rather than a cartoon depicting which
viruses 'won' and 'lost'*.

We fully agree that the results presentation in Figure 4 was unduly diagrammatic and lacked primary data (with the exception of the
two example gel images in Figure 4). This point
was also made by Reviewer #3. To address this, we have made use of the possibility
to include figure supplements by now including gel images of the other competition
assays so that relative fitness can be directly evaluated. We have, however, retained
the original Figure 4 as a summary of the
experimental data.

The use of restriction enzymes to differentiate E7 mutants is a widely used method and
its results can be made fully quantitative using appropriately calibrated controls. Had
the phenotypes been more subtle we agree that quantitative qRT-PCR would likely have
been necessary to discriminate between the fitness of viruses with modified dinucleotide
ratios. However, in the current study, outcomes were either elimination of one or other
of the competing viruses or a draw. These results are readily visualised as presented,
and we believe the investigation does not need more precise quantitation as suggested by
the reviewer.

*5) The last paragraph of the discussion uses 'data not shown' to make
its point. These data should be shown or the sentences deleted*.

We agree, and that part of the discussion has entirely been removed. We have also
removed reference to unpublished investigations of Theiler’s virus and influenza
A virus from the Introduction.

Reviewer #2

*1) The topic of dinucleotide biases in RNA viruses has been explored extensively
and most references are absent*.

We have now cited, in the Introduction, the Rima and Karlin studies that originally
noted the suppression of CpG and UpA dinucleotide frequencies in RNA viruses.

*2) The discussion on the mechanisms of attenuation and the association to the
innate immune response is not well elaborated. They mention the lack of association
to interferon response and the effect of C16. However, the results are not presented
in detail. I suggest the authors to strengthen their conclusions*.

That part of the Discussion was based on the results presented in the Atkinson et al.
paper from earlier in the year. We have modified that paragraph to clarify that we were
referring to this previous study.

Reviewer #3

*This study examines the role of nucleotide composition in RNA virus genomes. The
authors introduced a number of synonymous mutations into the EV71 genome to modify
the frequency of CpG and UpA dinucleotide (DN) or codon pair (CP). Their results
support the idea that DN frequencies determine virus replication, while neither DN
nor CP affected translation efficiency. They concluded that DN affect virus
replication because the nucleotide composition of the virus genome influences the
host-cell innate response to the virus*.

*The most interesting contributions of this study are: 1) they are able to
identify mutations that will affect CP without affecting CpG/UpA dinucleotide
composition and vice versa; (*[2]*) based on this
bioinformatics information they constructed mutants that change CP or DN composition
and experimentally evaluated the effect of synonymous mutation on virus replication.
Their conclusions are supported by the available results, and I think this study
represents an important contribution to the field because it seems to address a
molecular mechanism for virus genome nucleotide composition bias. However, I believe
that a more quantitative analysis of the competition experiment may be required to
determine the degree of correlation between DN composition and virus replication
fitness*.

*The authors choose competition assays to precisely analyze fitness between
different virus and with respect to WT. This is the correct experiment in my opinion,
however I believe the analysis of the experiment is somewhat casual and not very
quantitative, and therefore limits the value of this data.*
Figure 4
*presents the results using a differential restriction enzyme pattern to
distinguish between the two competing viruses, but they can only determine when one
virus is lost (no longer detected by the assay). I think that it will be a lot more
powerful to use digital PCR to precisely quantify the ratio between virus genomes in
the given competition assay. This will provide parameters that can then be fit in a
simple mathematical model to determine with more accuracy the correlation between DN
or CP and fitness, which at this point seems a bit circumstantial*.

This comment is related to that of Reviewer #1 and has been addressed above (the
limited number of observed outcomes of competition assays can be effectively
demonstrated through restriction enzyme analysis).

To address the issue of relative fitness further, we have repeated the experiment to
determine the replication kinetics of WT and mutants of E7 that was depicted in Figure 3. The re-formatting of data as histograms
with error bars now allows replication rates from all mutants at different time points
to be shown and these entirely back up the competition assay results.

*Similarly, it would be desirable to improve the quality of the in vitro
translation assay and quantify protein production to determine that, in this case,
there is little correlation between translation efficiency and fitness, as this is
one of the central claims of the study*.

We did indeed quantify translation of the individual proteins on the blot by
densitometry and presented the results in the form of a histogram in Figure 5–figure supplement 1. We additionally
used a translation efficiency metric based on this quantitation to analyse potential
associations with virus replication rate (Table 4) along with other variables (dinucleotide composition, CB bias, CAI, ENc and
G + C content).